# The Contribution of Host Cells to *Pneumocystis* Immunity: An Update

**DOI:** 10.3390/pathogens8020052

**Published:** 2019-04-19

**Authors:** Patricia Otieno-Odhiambo, Sean Wasserman, J. Claire Hoving

**Affiliations:** 1AFGrica Medical Mycology Research Unit, Institute of Infectious Diseases and Molecular Medicine, University of Cape Town, Cape Town 7925, South Africa; OTNPAT001@myuct.ac.za; 2Medical Research Council Centre for Medical Mycology at the University of Aberdeen, Institute of Medical Sciences, Foresterhill, Aberdeen AB25 2ZD, UK; 3Division of Immunology, Department of Pathology, University of Cape Town, Cape Town 7925, South Africa; 4Wellcome Centre for Infectious Diseases Research in Africa, Institute of Infectious Diseases and Molecular Medicine, University of Cape Town, Cape Town 7925, South Africa; sean.wasserman@uct.ac.za; 5Division of Infectious Diseases and HIV Medicine, Department of Medicine, University of Cape Town, Cape Town 7925, South Africa

**Keywords:** *Pneumocystis*, *Pneumocystis* life forms, alveolar macrophages, dendritic cells, lymphocytes

## Abstract

*Pneumocystis* is a ubiquitous atypical fungus that is distributed globally. The genus comprises morphologically similar but genetically heterogeneous species that have co-evolved with specific mammalian hosts as obligate intra-pulmonary pathogens. In humans, *Pneumocystis jirovecii* is the causative organism of Pneumocystis pneumonia (PCP) in immunocompromised individuals, a serious illness frequently leading to life-threatening respiratory failure. Initially observed in acquired immunodeficiency syndrome (AIDS) patients, PCP is increasingly observed in immunocompromised non-AIDS patients. The evolving epidemiology and persistently poor outcomes of this common infection will require new strategies for diagnosis and treatment. A deeper understanding of host immune responses and of the cells that mediate them will improve the chance of developing new treatment strategies. This brief review provides an update on recent studies on the role of host immunity against *Pneumocystis*.

## 1. Introduction

Sero-prevalence studies involving anti-*Pneumocystis* IgG in infants older than two years of age demonstrate evidence of near-universal primary infection with *Pneumocystis jirovecii* in early childhood, with clearance and transient colonisation throughout life [1]. Disease manifests in immunosuppressed individuals, either from reactivation of ‘latent’ infection acquired in childhood or through airborne transmission of new strains [1,2]. The incidence of human immunodeficiency virus (HIV)-associated Pneumocystis pneumonia (PCP), the sentinel opportunistic infection at the beginning of the AIDS epidemic, has declined dramatically in high-resource countries over the past three decades due to the introduction of potent combination antiretroviral therapy and effective chemoprophylaxis [3,4]. However, the epidemiology of PCP is shifting in these settings towards non-HIV-infected immunocompromised patients, with more widespread use of potent immunosuppressive therapy [5] in haematological malignancy, solid tumours, autoimmune conditions, and transplant recipients [6,7,8,9]. PCP incidence varies widely due to heterogeneity in study populations and designs but is reportedly as high as 15% in solid-organ transplant recipients [10] and over 18% in patients with solid tumours who are not on chemoprophylaxis [11]. HIV-negative PCP is associated with more rapid disease progression and worse outcomes than HIV-associated PCP [7,12], suggesting an influence of differential immune responses on clinical phenotype [13]. Case fatality is high, ranging between 20 and 67% [7,9,14,15]; this variability is related to the underlying condition [7,14] and severity of illness [9,15]. The prevalence of PCP in HIV-infected people of resource-limited countries remains difficult to ascertain, but it is increasingly recognised as an important cause of community-acquired pneumonia [16] and is possibly associated with a rising GDP. A recent systematic review found a prevalence of over 20% amongst inpatients presenting with respiratory symptoms in sub-Saharan Africa [17]. Outcomes are poor in these settings, with a pooled case fatality of 15% [17], reaching 60% in those admitted to ICU at one centre [18]. Improved diagnostics and treatment strategies are needed to improve patient prognosis in these settings. However, progress in the field is significantly hindered by the biotrophic nature of the *Pneumocystis* spp. This is a term adapted from plant fungal pathogens due to the dependence on their host for survival [19,20]. Therefore, *Pneumocystis* cannot be propagated in culture. Furthermore, *Pneumocystis* has strict host species specificity, with *P. jirovecii* infecting humans, *Pneumocystis carinii* infecting rats and *Pneumocystis murina* infecting mice. The substantial variation in the genomes of rodent and human *Pneumocystis* spp. suggests there may be clinically relevant differences between animal models of PCP and human disease. However, studies using animal models have provided valuable insight into understanding host immune mechanisms during infection. Ma and colleagues have elegantly summarised advances in molecular methods that have led to a better understanding of the biology and epidemiology of *Pneumocystis* infection [2,21]. Since being identified as a fungus using 16S rRNA gene analysis, *Pneumocystis* taxonomic classification has improved, and it is now classified in the monophyletic subphylum Taphrinomycotina. Molecular typing methods which identify *Pneumocystis* strain variation have led to an improved understanding of drug resistance and the pathogenesis of *Pneumocystis* species [2]. Furthermore, a recent study suggested that the virulence of *P. jirovecii* strains or genetic polymorphisms in humans could be responsible for treatment failure and high mortality in 12 Indian patients [22]. New studies continue to dissect host–pathogen interactions, with the goal of identifying key immune components capable of specifically targeting *Pneumocystis*. These include recognition of various life forms of *Pneumocystis* species by host dendritic cells (DCs) and alveolar macrophages and the role of the innate immune system in triggering the appropriate adaptive response. Ultimately, it is an effective adaptive CD4^+^ T cell response that is pivotal to *Pneumocystis* clearance. However, the specific nature of this response, and the delineation of CD4^+^ T helper cell subsets, Th1, Th2 and Th17, is a subject of continuous investigation. Therefore, this brief review aims to provide an update on recent developments on the host immune response to *Pneumocystis* species. 

## 2. *Pneumocystis* Life Forms and Host Cells 

Important in understanding the host immune response to *Pneumocystis* spp. is characterising the life cycle stages during infection and how these influence the host response. The *Pneumocystis* lifecycle has an infective ascus form and an asexual trophic form. While *Pneumocystis* species have developed unique mechanisms which allow them to evade immune recognition and persist in the host, the life form stages also elicit distinct immune responses which influence the outcome of disease [23,24]. The ascus form has up to eight ascospores and a cell wall consisting of the major surface glycoproteins (MSG), β-1,3 glucan and β-1,6 glucan. β-1,3 glucan is the most abundant and is the primary pro-inflammatory factor. Like the ascus form, the asexual trophic form expresses MSG but not β-glucans [25]. Asci β-glucan serves as a pathogen-associated molecular pattern (PAMP) that is recognised by pattern recognition receptors (PRRs) on phagocytic cells. This recognition is essential in initiating an appropriate CD4^+^ T cell response to drive *Pneumocystis* clearance, as shown in Figure 1. Because of the lack of β-glucans, the trophic form is poorly recognised by phagocytic cells, resulting in an insufficient CD4^+^T cell response. It is commonly accepted that multiple types of phagocytic cells contribute to T cell priming in response to *Pneumocystis*. However, increasing evidence suggests that the life stage form of *Pneumocystis* also influences the host immune response [24,26]. Specifically, Evans and colleagues proposed that asci play a primary role in inducing the host immune response by stimulating interferon gamma (IFN-γ) production by CD4^+^ T cells. In contrast, the trophic form hinders dendritic cell proinflammatory cytokine production elicited by *β*-1,3-glucan, thereby suppressing the immune response. In fact, Evans et al. proposed that suppression could indeed be beneficial for immunocompromised hosts in that the lung pathology associated with PCP is reduced [24,26]. Furthermore, T cells primed in the presence of the trophic form can effectively mediate the clearance of both trophic and ascus forms [27]. The majority of studies have described the role of immune cells against *Pneumocystis*. Recently, a study by Kottom and colleagues provided evidence for the role of airway epithelial cells in binding and recognising microorganisms to initiate the relevant signalling pathways [28]. Here, they showed that the trophic form of *Pneumocystis* binds to alveolar epithelial cells, aiding proliferation, but that both asci and trophs mediate a cytokine response including IL-8, IL-6, TNFα and Macrophage Inflammatory protein 2 (MIP-2). As future studies discriminate the host immune response on the basis of specific life forms of *Pneumocystis*, it will be interesting to dissect the role this plays in resolving infection and potentially identify underlying mechanisms which allow *Pneumocystis* to go undetected during host colonisation. 

Significant progress has been made in understanding host immune responses to *Pneumocystis* through animal and human studies. The fact that *Pneumocystis* has multiple life stage forms influences the effectiveness of immune recognition and subsequent CD4^+^ T cell responses. Th1, Th2 and Th17 are implicated in *Pneumocystis* clearance. However, Th2 and Th17 are also associated with pathology, including mucin (Muc5ac) production. B cells are key in CD4^+^ T cell priming and as effector cells. M1 = classically activated macrophages, M2 = alternatively activated macrophages, *Pc = Pneumocystis*, rh = recombinant human, Th = T helper [24,29,30,31,32,33,34]

## 3. Dendritic Cells

Dendritic cells are professional antigen-presenting cells which elicit effector functions in the lung. Activated DCs produce cytokines and migrate to the draining lymph node where they activate T cell responses to antigens. This activation is important in the immunity to *Pneumocystis*, however limited studies have investigated the role it plays in initiating effective adaptive immune responses. Fas ligand (FasL) is important in DC activation and in regulating IL-1β produced by DCs [35]. Carmona et al. described the role of FasL in DC activation by *Pneumocystis*-derived β-glucans. Here, they demonstrated that β-glucans do indeed activate DCs through the FasL mechanism and Dectin-1 receptor, resulting in an increase in the expression of co-stimulatory molecules and subsequent Th1 polarisation [35]. To further investigate the role of β-glucan-driven DC activation, Carmona et al. found that human DCs stimulated by *Pneumocystis*-derived β-glucans interact with lymphocytes to produce IL-17. This is mediated by glycosphingolipids, downstream signalling molecules, [36]. In contrast, MSG seems not to be involved in DC activation. This was shown by Sassi et al., using RNA expression, cytokine production (including TNF-α and IFN-γ) and co-stimulatory molecule expression (such as CD40, CD80 and CD86) compared with Lipopolysaccharide (LPS)-stimulated controls [37]. In summary, these studies suggest that DCs play a significant role in initiating immune responses to *Pneumocystis*. These responses are driven by β-glucans rather than MSG, and the trophic form may actually hinder effective DC activation, resulting in impaired CD4^+^ T cell responses. 

## 4. Alveolar Macrophages

Alveolar macrophages (AMs) are key resident effector cells of alveolar spaces and critical in the clearance of lung microbes as the first line of defence. AMs are important for the recognition, phagocytosis and destruction of lung pathogens, including *Pneumocystis*. To achieve this, AMs have to be activated by host cytokines, such as IFN-γ, TNFα and granulocyte macrophage colony stimulating factor (GM-CSF), to maximise their fungicidal activity. Upon activation, AMs destroy *Pneumocystis* through the production of reactive oxygen and nitrogen species. The depletion of AMs in the lung thus results in impaired *Pneumocystis* clearance. Furthermore, evasion of AMs enhances the survival of *Pneumocystis* in the alveolar spaces [38]. Undeniably, AMs are critical in *Pneumocystis* clearance. However, recent studies investigating the response of macrophages to *Pneumocystis* infection suggested that the type of macrophage activation could also be significant in, and affect the level of clearance. The activation status of macrophages is categorised as either classical (M1) or alternative (M2). It is widely accepted that M1 macrophages are effective fungicidal cells, whereas M2 macrophages play an anti-inflammatory role [39]. A study dissecting the role of M1/M2 macrophages in *Pneumocystis* infection found that the host immune status plays a significant role [40]. Here, immunosuppressed rats exhibited a predominantly M1 phenotype, which resulted in defective *Pneumocystis* clearance. In contrast, the immunocompetent rats, exhibiting M2 macrophages, cleared the *Pneumocystis* infection effectively. Treating immunosuppressed rats with M2 macrophages resulted in *Pneumocystis* clearance and reduced inflammation [40]. Furthermore, Bhagwat et al. showed that intrinsic programming of AMs confers protective innate immunity against *Pneumocystis* infection [41]. Essentially, they showed that mice infected with Friend Leukemia Virus B (FVB) inherently have M2 macrophages which clear *Pneumocystis* infection before the onset of adaptive immunity. Here, the adoptive transfer of FVB M2 macrophages to immunocompromised RAG2^-/-^ mice conferred protection to *Pneumocystis* [41]. Collectively, these studies show that macrophage polarization represents a potential treatment model to improve *Pneumocystis*-associated immuno-pathogenesis.

## 5. Lymphocytes: CD4^+^ T Cells, CD8^+^ T Cells and B Cells

CD4^+^ T cells are crucial for *Pneumocystis* clearance. This has been demonstrated by the use of immunodeficient animal models, where loss of CD4^+^ T cells renders the animals susceptible to *Pneumocystis* infection. Furthermore, PCP is observed in patients with CD4^+^ T cell counts lower than 200 cells/mm^3^. CD4^+^ T cells confer memory cell functions that coordinate inflammatory responses in the host by recruiting and activating effector cells, like macrophages [23,24]. However, the contribution of specific CD4^+^ T cell subsets needs further investigation. In mice, *Pneumocystis* activates T helper cells, namely, Th1, Th2 and Th17 responses. Th17 responses are thought to play a role in *Pneumocystis* clearance, with selected studies suggesting the recruitment of CD4^+^ T cells producing interleukin 17A (IL-17A) in the lungs of infected animals. Furthermore, neutralization of IL-17A or its counterpart IL-23 results in a significant increase in *Pneumocystis* lung burden at later stages of infection. In addition, mice lacking nuclear factor (NF)-κβ signalling within the lung epithelium exhibit impaired fungal clearance that is associated with reduced lung IL-17^+^ CD4^+^ T cells [29,38,39,40,41,42,43]. Interestingly, Th17 cells are also thought to produce IL-9 [44]; how this influences Th17 responses to *Pneumocystis* remains to be determined. Investigations by Li et al. suggested that, despite *Pneumocystis* clearance in wild-type (WT) and IL-9-deficient mice, IL-9 deficiency may actually lower lung burden and promote pulmonary Th17 responses in the early stages of infection [45]. Th1 and Th2 have also been implicated in host protection against *Pneumocystis*. A study by Kolls et al. showed that in the absence of CD4^+^ T cells, artificial activation of IFN-γ reinstates control of infection. This suggests a role for Th1 responses in conferring protection against *Pneumocystis* infection [46]. Similarly, Th2-associated responses and M2 macrophages are linked to protection against *Pneumocystis* and fungicidal activity [40,47]. In addition, Eddens and colleagues provided convincing evidence for the contribution of eosinophils in *Pneumocystis* clearance. In their study, RAG-1-deficient mice treated with plasmid-derived IL-5 had increased eosinophilia in the lung, with reduced *Pneumocystis* burden [48]. The authors proposed that it is the early response of CD4^+^ T cells that recruits eosinophils. In contrast, a Th2 immune response to *Pneumocystis* has been shown to induce pathology. That is, Th2 responses lead to an asthma-like pathology in immunocompetent individuals, similar to house dust mite-induced asthma [30]. Similarly, in rodents, *Pneumocystis* induced a Th2–type inflammation and airway remodelling [31]. Furthermore, the slow rise in the mucin Muc5ac during primary *Pneumocystis* infection suggested an ongoing Th2-type immune response linked to airway sensitization against this fungus [49]. Recently, the mucin Muc5b was shown to be increased earlier and more abundantly than Muc5ac during primary infection in rats [32]. This suggested an acute defensive response against *Pneumocystis*, in addition to the central effector role of Muc5ac in mediating Th2-type allergic inflammation [32]. The importance of early host responses in controlling infection is further highlighted in previous studies, such as the expression of ClCa3 on airway goblet cells, which drives mucin production after infection [50] and the fact that *P. carinii* can bind to rat epithelial cells [28]. In summary, these studies indicated a fundamental role for CD4^+^ T cell-mediated responses to *Pneumocystis* infection and also highlighted the participation of host airway cells in the immune response against *Pneumocystis*. 

CD8^+^ T cells are thought to work in conjunction with CD4^+^ T cells to elicit an effective immune response against *Pneumocystis*. The role of CD8^+^ T cells in the absence of CD4^+^ T cells remains controversial, with reports of both protective and detrimental responses. Early studies by Kolls et al. showed that IFN-γ resulted in an increase in IFN-γ^+^ CD8^+^ T cells, which enhanced effective clearance of *Pneumocystis* infection in the absence of CD4^+^ T cells [46]. In addition, in vitro studies by Mc Allister et al. showed that IFN-γ-stimulated CD8^+^ T cells derived from CD4^+^-depleted *Pneumocystis*-infected mice were able to enhance macrophage-mediated killing [33]. Recently, a study by Ruan et al. demonstrated that the continual administration of recombinant human IL-7 (rhIL-7) to CD4^+^ depleted mice resulted in an increase in both CD8^+^ T cells and CD8^+^ memory T cells recruited to the lungs. This was correlated to *Pneumocystis* clearance. Additionally, they demonstrated that rhIL-7 administration enhanced lymphocyte function, including activation, proliferation and IFN-γ expression, and reduced cell death at the site of infection [34]. Human IL-7 has been shown to bind and signal via the murine IL-7 receptor, and the authors therefore speculated that the response is due to the increment of IFN-γ-secreting CD8^+^ T cells, that are cytotoxic to *P. murina*, at the site of infection. On the other hand, a study by Gigliotti et al. indicated no difference in the lung burden between CD4^+^ and CD8^+^ T cell-depleted mice, suggesting that the role of CD8^+^ T cells in *P. murina* clearance is questionable [51]. Further investigation of the role of CD8^+^ T cells in *Pneumocystis* infection focussed on secondary immune responses. De la Rua and colleagues suggested that secondary immune responses involve CD8^+^ T cells and alveolar macrophages. Briefly, CD8^+^ T cells and macrophage depletion prior to a secondary infection significantly impairs *Pneumocystis* clearance compared with CD4^+^ depleted mice, which clears infection within 48 h, comparable to immunocompetent mice. Furthermore, the loss of alveolar macrophages resulted in an increase in IFN-γ^+^ CD8^+^ T cells [52]. These studies showed evidence of the role of CD8^+^ T cells in *Pneumocystis* clearance. The role of CD8^+^ T cells remains controversial, and one would need to consider other potential causes for the conflicting results. These could be influenced by, for example, the life-cycle stages present in lung homogenates from infected lungs and the method used for immunosuppressing experimental animals.

Susceptibility to *Pneumocystis* is also influenced by B cells, which play a dual role in antibody production and antigen presentation [53]. Firstly, B cells are important in T cell priming during the primary stage of *Pneumocystis* infection. Lund et al. demonstrated that cognate B–T cell interactions are important for the generation of both effector and memory CD4^+^ T cells in *Pneumocystis* infection [54]. Opata et al. went further to show that these B cells play an important role in early priming of CD4^+^ T cells [55]. Both studies highlighted the significance of MHC class II expression in B cells, where the lack thereof results in defective CD4^+^ T cell immunity and susceptibility to *Pneumocystis*. Immunoglobulin M (IgM) is key in fungal recognition by dendritic cells, B cell isotype class-switching and Th2 and Th17 promotion of differentiation. CD20 is a membrane-bound surface molecule expressed on B cells and is important in the development and differentiation of B cells into plasma cells. Mice lacking IgM and those treated with anti-CD20 exhibit CD4^+^ T cell priming defects [56,57]. On the contrary, patients with B cell disorders such as x-linked agammaglobulinemia and common variable immunodeficiency (CVID) are reported to rarely present with PCP, suggesting a B cell-independent CD4^+^ T cell priming mechanism. In terms of antibody production, Gigliotti et al. [58] and Zheng et al. [59] demonstrated that the passive transfer of immune serum averts *Pneumocystis* infection. These studies suggested that B cells play a role in CD4^+^ T cell priming and as effector cells. 

Taken together, *Pneumocystis* infection activates Th1, Th2 and Th17 responses which are implicated in the immune response to *Pneumocystis*. However, Th2 and Th17 responses can also induce pathology in competent hosts. CD8^+^ T cells play a role in secondary immune responses, and B cells are important antigen-presenting and effector cells in *Pneumocystis* infection.

## 6. Conclusions

Significant progress has been made in understanding the host immune response to *Pneumocystis*. Dendritic cells play a central role in initiating these immune responses, which are influenced by β-glucan recognition from specific *Pneumocystis* life forms. The role of alveolar macrophages as key effector cells in clearing *Pneumocystis* is undisputed. However, the macrophage activation status seems to be of importance and is dependent on the immune status of the host. M1 macrophages in an immunocompromised host are defective, and adoptive transfer of M2 macrophages assists in clearance. CD4^+^ T cells are essential in controlling *Pneumocystis* infection, but the specific T helper cell subset that is indispensable is yet to be determined. This is likely due to potential compensatory mechanisms that control infection in the absence of specific subsets. Furthermore, both Th2 and Th17 responses have been associated with *Pneumocystis*-driven pathology. CD8^+^ T cells play a role in memory recall responses in the absence of CD4^+^ T cells, and B cells are important in CD4^+^ T cell priming and as effector cells. Understanding the role host cells play in the protection against *Pneumocystis* infection and identifying components that contribute to disease progression could lead to developing better treatment options. New treatment strategies will prove essential, on the basis of reports of the increase in PCP cases and drug resistance. 

## Figures and Tables

**Figure 1 pathogens-08-00052-f001:**
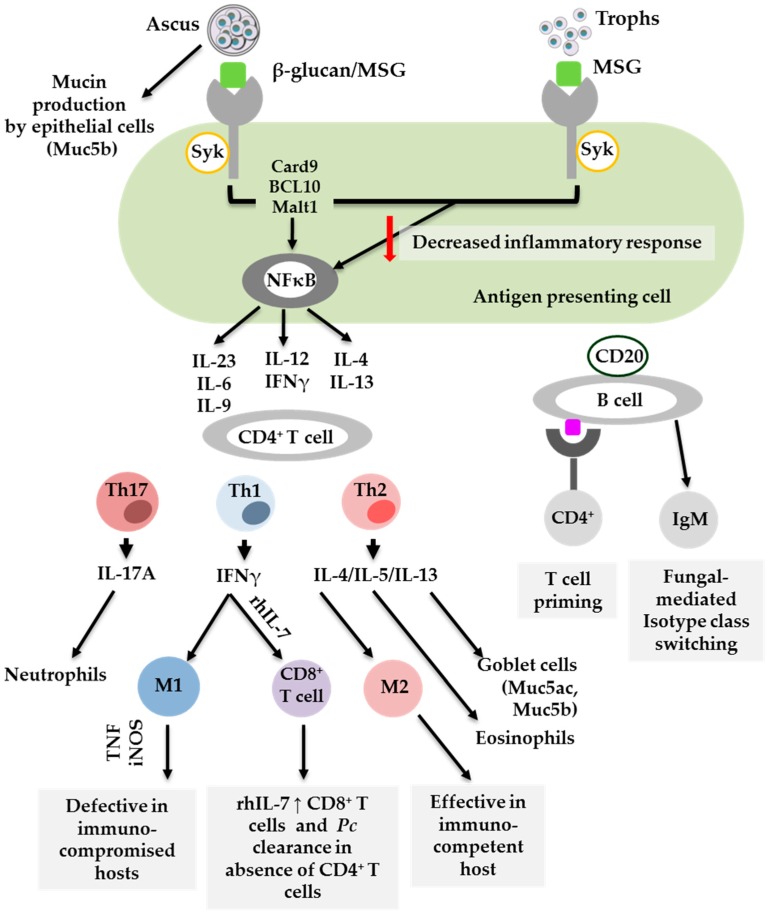
Cells which contribute to the host immune response to *Pneumocystis*.

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
