# Peer review of "The Contribution of Host Cells to Pneumocystis Immunity: An Update"

_pathogens, 2019, doi:10.3390/pathogens8020052_

Round 1
Reviewer 1 Report
The authors, internationally recognized in the topic of the host response to Pneumocystis write an interesting review about the contribution of host cells to Pneumocystis immunity. The article is well written. I hope the following comments will improve the quality of the manuscript.
minor: the first sentence of the manuscript lacks precision as refers to the documentation of near-universal primary infection in early childhood by sero-prevalence studies.
The sentence should be improved to indicate that seroprevalence studies have demonstrated near-universal prevalence of anti-Pneumocystis IgG antibodies in infants older than 2 years of age suggesting primary infection is common in early childhood. The sentence as written in the manuscript is misleading because IgG antibodies are of no diagnostic value in Pneumocystis infection.
Major.
The authors do not refer to the contribution of the airway epithelium to the host response. They should reference the important article by Hernandez-Novoa and Kovacs (J Leuk biol 2008) documenting activation of host gene responses by microarray and hyperexpression of CLCA1 suggesting goblet cell airway epithelium responses, and the work of the Vargas' laboratory of MUC5AC mucin increase in infant autopsy samples reported in Clin Infect Dis 2013 and the further documentation of Th2 type airway pathology in rats in the Journal of Pathology 2018 where a Th2 response with eosinophils was documented as in the work by Eddens and Kolls cited as reference 42. The reference by Rojas et. al. documents MUC5B increase in infant autopsy samples. This work overall together with the recent report by Kottom et al (ref 26) well referenced by the authors suggest that host airway cells participate in the immune response against Pneumocystis.
The authors should consider including goblet cells, MUC5AC and MUC5B in Figure 1.
Author Response
Reviewer 1:
Minor: the first sentence of the manuscript lacks precision as refers to the documentation of near-universal primary infection in early childhood by sero-prevalence studies.
The sentence should be improved to indicate that seroprevalence studies have demonstrated near-universal prevalence of anti-Pneumocystis IgG antibodies in infants older than 2 years of age suggesting primary infection is common in early childhood. The sentence as written in the manuscript is misleading because IgG antibodies are of no diagnostic value in Pneumocystis infection.
Response
We have amended this sentence as follows:
“Sero-prevalence studies involving anti-Pneumocystis IgG in infants older than two years of age demonstrates have demonstrated evidence of near-universal primary infection with Pneumocystis jirovecii in early childhood, with clearance and transient colonization throughout life.”
Major:
The authors do not refer to the contribution of the airway epithelium to the host response. They should reference the important article by Hernandez-Novoa and Kovacs (J Leuk biol 2008) documenting activation of host gene responses by microarray and hyperexpression of CLCA1 suggesting goblet cell airway epithelium responses, and the work of the Vargas' laboratory of MUC5AC mucin increase in infant autopsy samples reported in Clin Infect Dis 2013 and the further documentation of Th2 type airway pathology in rats in the Journal of Pathology 2018 where a Th2 response with eosinophils was documented as in the work by Eddens and Kolls cited as reference 42. The reference by Rojas et. al. documents MUC5B increase in infant autopsy samples. This work overall together with the recent report by Kottom et al (ref 26) well referenced by the authors suggest that host airway cells participate in the immune response against Pneumocystis.
The authors should consider including goblet cells, MUC5AC and MUC5B in Figure 1.
Response
Thank you for these additional suggestions. We hope to have included these points by adding the following:
Similarly, Th2 associated responses and M2 macrophages are linked to protection against Pneumocystis and fungicidal activity [34,42]. In addition, Eddens and colleagues provide convincing evidence for the contribution of eosinophils in Pneumocystis clearance. In their study RAG-1-deficient mice treated with plasmid derived IL-5 had increased eosinophilia in the lung with reduced Pneumocystis burden [43]. The authors propose that it is the early response of CD4+ T cells that recruits eosinophils. In contrast, a Th2 immune response to Pneumocystis has been shown to induce pathology. That is, Th2 responses lead to asthma like pathology in immunocompetent individuals almost similar to house dust mite induced asthma [44]. Similarly in rodents, Pneumocystis induces a Th2–type inflammation and airway remodelling [45]. Furthermore, the slow rise in the mucin Muc5ac, during primary Pneumocystis infection suggests an ongoing Th2-type immune response linked to airway sensitization against this fungus [46]. Recently, the mucin Muc5b was shown to be increased earlier and more abundantly than Muc5ac during primary infection in rats [47]. This suggests an acute defensive response against Pneumocystis in addition to the central effector role of Muc5ac in mediating Th2-type allergic inflammation [47]. The importance of early host responses in controlling infection are further highlighted in previous studies such as, the expression of ClCa3 on airway goblet cells, which drives mucin production after infection [48] and the fact that Pneumocystis carinii can bind to rat epithelial cells [28]. In summary, these studies indicate a fundamental role for CD4+ T cell mediated responses to Pneumocystis infection and also highlight the participation of host airway cells in the immune response against Pneumocystis”.
Associated new references:
[44] Eddens T, Elsegeiny W, Nelson MP, Horne W, Campfield BT, Steele C, Kolls JK. Eosinophils Contribute to Early Clearance of Pneumocystis murina Infection. J Immunol. 2015 Jul 1;195(1):185-93. doi: 10.4049/jimmunol.1403162.
[45] Iturra PA, Rojas DA, Pérez FJ, Méndez A, Ponce CA, Bonilla P, Bustamante R, Rodríguez H, Beltrán CJ, Vargas SL. Progression of Type 2 Helper T Cell–Type Inflammation and Airway Remodeling in a Rodent Model of Naturally Acquired Subclinical Primary Pneumocystis Infection. The American Journal of Pathology, 188,2, 2018, 417-431
[46] Vargas SL1, Ponce CA, Gallo M, Pérez F, Astorga JF, Bustamante R, Chabé M, Durand-Joly I, Iturra P, Miller RF, Aliouat el M, Dei-Cas E. Near-universal prevalence of Pneumocystis and associated increase in mucus in the lungs of infants with sudden unexpected death. Clin Infect Dis. 2013 Jan;56(2):171-9. doi: 10.1093/cid/cis870. Epub 2012 Oct 16.
[48] Hernandez-Novoa B, Bishop L, Logun C, Munson PJ, Elnekave E, Rangel ZG, Barb J, Danner RL, Kovacs JA. Immune responses to Pneumocystis murina are robust in healthy mice but largely absent in CD40 ligand-deficient mice. J Leukoc Biol. 2008 Aug;84(2):420-30
We have included additional data into Figure 1
Reviewer 2 Report
This is a well -written summary of the current thought on the immunological responses of mammalian hosts to fungal species in the genus, Pneumocystis. It is also a welcome review, as there have been many papers published in the last few years, dissecting these immune responses.
I have both general and specific comments:
General
The authors should note the existence of species specificity for these fungi early in the review. This is of extreme importance because so much of the information we have has been performed in mice and rodents, which may or may not have relevance to responses in humans. This is especially true in light of the many ways that the rodents have been immunosuppressed, which can, again, affect outcome. I will come back to this concept in the Detailed Comments, below.
It is beyond doubt that Pneumocystis are Fungi. Thus, antiquated nomenclature such as "cyst" or "cyst-like" used when there was still a dilemma regarding its protozoan or fungal identity, need to be stricken from modern manuscripts.
In the first paragraph, reactivation of "latent infection" is brought up. This is somewhat misleading, since colonized individuals can develop full blown PCP upon immune suppression, but this term has been used to refer to the infection that was acquired during infancy. We now understand that humans can be transiently colonized throughout our life time, with clearance in immune intact individuals. The authors need to clarify.
Detailed
Biotrophy is a novel concept that describes a plant fungal pathogen and its host and was adapted to the unusual relationship of Pneumocystis spp. to its mammalian hosts. It is worth a mention and the references to its authors, Cushion 2007 and Hauser 2014.
The same is true for the term "biphasic". See Liu GS et al. BMC Syst Biol, 2018 (Page 2)
It should be noted, "the sexual cystic form" (should be ascus) does not contain simply "several nuclei" but rather up to 8 ascospores. (Page 2)
The discussion on B-glucans should note that B-1,3-D-glucan is in dramatically more abundance than B-1,6 glucan and is the primary pro-inflammatory factor. (Page 2)
Fig. 1- change "cyst" to ascus.
As I remarked above, Pneumocystis species are specific for a mammalian host. Thus , one does not find Pneumocystis carinii (the species that infects rats) in mice. P. murina are found exclusively in mice; and P. jirovecii infect only humans. The study by Ruan that was cited on Page 5, describes the use of human recombinant IL-7 that was administered to mice, which brings up the question of what the response was reacting to- the human protein pehaps?
Gigliotti et al is cited on the same page, with "Pneumocystis carinii" in its title when in fact, they used mice and P. murina in these studies. This lack of compliance to formally named species leads to confusion in the literature.
The CD8+ story is indeed confusing. James Beck identified the role of these cells in clearance in a 1996 paper.
It should also be noted that many studies, until recently, used homogenates of infected lungs to conduct immunological studies, with no reference to the number of asci or trophs in these preparations nor did they appreciate the differential response that may be due to the life cycle stage. I recommend that the authors make note of that which is likely an important factor in contradictory results. Another reason for mixed results is likely due to the animal model used and its method of immunosuppression. It would be a service to readers to understand this.
Author Response
Reviewer 2:
General
The authors should note the existence of species specificity for these fungi early in the review. This is of extreme importance because so much of the information we have has been performed in mice and rodents, which may or may not have relevance to responses in humans. This is especially true in light of the many ways that the rodents have been immunosuppressed, which can, again, affect outcome. I will come back to this concept in the Detailed Comments, below.
Response
We have included the following to highlight these points:
“Furthermore, Pneumocystis has strict host species specificity with Pneumocystis jirovecii infecting humans, Pneumocystis carinii in rats and Pneumocystis murina in mice. The substantial variation in the genomes of rodent and human Pneumocystis spp. suggests there may be clinically relevant differences between animal models of PCP and human disease. However, studies using animal models have provided valuable insight into understanding host immune mechanisms during infection. Ma and colleagues have elegantly summarized advances in molecular methods that have led to a better understanding of the biology and epidemiology of Pneumocystis infection [2, 21].”
It is beyond doubt that Pneumocystis are Fungi. Thus, antiquated nomenclature such as "cyst" or "cyst-like" used when there was still a dilemma regarding its protozoan or fungal identity, need to be stricken from modern manuscripts.
Response
Thank you for pointing this out, we have amended this through the manuscript.
In the first paragraph, reactivation of "latent infection" is brought up. This is somewhat misleading, since colonized individuals can develop full blown PCP upon immune suppression, but this term has been used to refer to the infection that was acquired during infancy. We now understand that humans can be transiently colonized throughout our life time, with clearance in immune intact individuals. The authors need to clarify.
Response
We have rephrased this as follows, “Disease manifests in immunosuppressed individuals either from reactivation of ‘latent’ infection acquired in childhood or through airborne transmission of new strains.”
Detailed
Biotrophy is a novel concept that describes a plant fungal pathogen and its host and was adapted to the unusual relationship of Pneumocystis spp. to its mammalian hosts. It is worth a mention and the references to its authors, Cushion 2007 and Hauser 2014.
Response
We thank the reviewer for this insightful comment. We have included the following:
“However, progress in the field is significantly hindered by the Pneumocystis is biotrophic nature of the Pneumocystis spp. This is a term adapted from plant fungal pathogens due to the dependence on their host for survival [19, 20]”.
The same is true for the term "biphasic". See Liu GS et al. BMC Syst Biol, 2018 (Page 2)
Response
To avoid confusion we have removed this term and amended the text as follows:
“The Pneumocystis lifecycle has an infective ascus form and an asexual trophic form.”
It should be noted, "the sexual cystic form" (should be ascus) does not contain simply "several nuclei" but rather up to 8 ascospores. (Page 2)
Response
This has been rephrased as follows
“The ascus form has up to 8 ascospores and a cell wall consisting of the major surface glycoprotein (MSG), and β-1,3 glucan and β-1,6 glucan.”
The discussion on B-glucans should note that B-1,3-D-glucan is in dramatically more abundance than B-1,6 glucan and is the primary pro-inflammatory factor. (Page 2)
Response
This has been rephrased as follows:
“β-1, 3 glucan is the most abundant and the primary pro-inflammatory factor.”
Fig. 1- change "cyst" to ascus.
Response
Amended
As I remarked above, Pneumocystis species are specific for a mammalian host. Thus, one does not find Pneumocystis carinii (the species that infects rats) in mice. P. murina are found exclusively in mice; and P. jirovecii infect only humans.
Response
As mentioned above we have amended as follows:
“Furthermore, Pneumocystis has strict host species specificity with Pneumocystis jirovecii infecting humans, Pneumocystis carinii in rats and Pneumocystis murina in mice. The substantial variation in the genomes of rodent and human Pneumocystis spp. suggests there may be clinically relevant differences between animal models of PCP and human disease. However, studies using animal models have provided valuable insight into understanding host immune mechanisms during infection. Ma and colleagues have elegantly summarized advances in molecular methods that have led to a better understanding of the biology and epidemiology of Pneumocystis infection [2, 19].”
The study by Ruan that was cited on Page 5, describes the use of human recombinant IL-7 that was administered to mice, which brings up the question of what the response was reacting to- the human protein pehaps?
Response
We appreciate this response and have added the following in support of the response being mediated by the signalling of IL-7.
”Human IL-7 has been shown to bind and signal via the murine IL-7 receptor, and the authors therefore speculate that the response is due to the increment of IFN-γ secreting CD8+ T cells at the site of infection that are cytotoxic to P. murina.”
Gigliotti et al is cited on the same page, with "Pneumocystis carinii" in its title when in fact, they used mice and P. murina in these studies. This lack of compliance to formally named species leads to confusion in the literature.
Response
Thank you for this comment. We have included P. murina to clarify the species used for the studies.
The CD8+ story is indeed confusing. James Beck identified the role of these cells in clearance in a 1996 paper.
It should also be noted that many studies, until recently, used homogenates of infected lungs to conduct immunological studies, with no reference to the number of asci or trophs in these preparations nor did they appreciate the differential response that may be due to the life cycle stage. I recommend that the authors make note of that which is likely an important factor in contradictory results. Another reason for mixed results is likely due to the animal model used and its method of immunosuppression. It would be a service to readers to understand this.
Response
We hope to have addressed this concern as follows:
“The role of CD8+ T cells remains controversial and one would need to consider other potential causes for conflicting results. These could be influenced by, for example the life-cycle stages present in lung homogenates from infected lungs and the method used for immunosuppressing experimental animals”.
Round 2
Reviewer 1 Report
The authors have responded to this reviewer comments.
Minor issues:
In figure 1:
Would add "Muc5b" after Muc5ac below Goblet cells. This is documented in reference 47.
In figure 1 legend:
Include reference "38" to strengthen the connection between NFkB and Th17.
Include reference "45" that contributes to understanding pathology and Muc5ac increase.
Line 194: A reference typo and space need to be corrected after the word "asthma".
Line 201: "]." is missing.
Author Response
Minor issues:
In figure 1:
Would add "Muc5b" after Muc5ac below Goblet cells. This is documented in reference 47.
In figure 1 legend:
Include reference "38" to strengthen the connection between NFkB and Th17.
Include reference "45" that contributes to understanding pathology and Muc5ac increase.
Line 194: A reference typo and space need to be corrected after the word "asthma".
Line 201: "]." is missing.
Response:
We thank the reviewer for pointing out these errors and have corrected all of them as suggested.